# Measuring Urban Infrastructure Resilience via Pressure-State-Response Framework in Four Chinese Municipalities

**Min Chen** [1], **Yu Jiang** [1,*], **Endong Wang** [2], **Yi Wang** [1,*] and **Jun Zhang** [1]

1   School of Transportation and Civil Engineering, Nantong University, Nantong 226000, China; chen.min@ntu.edu.cn (M.C.); 13951411616@139.com (J.Z.)
2   Sustainable Construction Engineering Program, State University of New York, Syracuse, NY 13210, USA; ewang01@esf.edu
*   Correspondence: 1933310002@stmail.ntu.edu.cn (Y.J.); wang12yi@ntu.edu.cn (Y.W.)

**Abstract:** Urban infrastructure (UI), subject to ever-increasing stresses from artificial activities of human beings and natural disasters due to climate change, assumes a key role in modern cities for maintaining their functional operations. Therefore, understanding UI resilience turns essential. Based on the Pressure-State-Response (PSR) model, this paper built a comprehensive evaluation index system for urban infrastructure resilience evaluation. Four municipalities, including Beijing, Tianjin, Shanghai, and Chongqing in China, were selected for the case study, given their specific significance in terms of geographical location and urban infrastructure scale. Temporal differences of UI resilience in those four cities during 2002–2018 were explored. The results showed that: (1) The various stages of PSR relative importance for the urban infrastructure resilience development in the four cities were different. The infrastructure status, primarily resource environmental benefit, had the most significant effect on urban infrastructure resilience, accounting for 38.73%. (2) While Shanghai ranked first, the levels of urban infrastructure resilience in four cities were generally poor in 2002–2018 with continuously low resilience. (3) Significant differences were found in the resilience levels associated with the three stages of pressure, state and response failing to form a positive development cycle, with the poorest pressure resilience. This paper puts forward some recommendations for providing scientific support for urban resilient infrastructure development in four municipalities in China.

**Keywords:** urban infrastructure; resilience; pressure-state-response; Chinese Municipalities; temporal differences

## 1. Introduction

With the rapid growth of industrialization and urbanization, Chinese cities have gradually become the leading carriers for significant populations to settle. As the population increases, the city size has been expanding quickly. In China, the urban population quintupled from 170 million in 1978 to more than 850 million in 2020. Meanwhile, the urbanization rate has nearly quadrupled during that period, from 18% in 1978 to over 60% in 2020, and is expected to reach 75% or even 80% by 2035 [1]. In urbanization, the Chinese government has maintained the growth rate of economic investment in UI development has been maintained at around 20% by Chinese government. The scale of UI has increased substantially, establishing relatively complete urban infrastructure systems in cities. However, "urban diseases" have been increasingly emerging, especially in metropolis [2], such as traffic congestion, urban pollution, and poor disaster resilience, indicating that infrastructures' carrying capacity lags far behind urban development speed. In the traditional sense, UI is the general name of engineering infrastructure and social infrastructure, and it is necessary for urban operation and development. Engineering infrastructure is generally divided into six systems (transportation, water and drainage, communication, energy source supply, urban environment, and disaster prevention) according to the "Standard for Basic Terminology of Urban Planning" (GB/T50280-98). These six engineering infrastructures serve

people but also serve other infrastructures, and jointly constitute an open, complex and dynamic system. In this case, this paper defined UI as the engineering infrastructure. As an essential material foundation for the functional operation and healthy development of a city, UI plays a vital role in satisfying the living conditions of citizens, enhancing total carrying capacity, and improving urban operational efficiency. Once the infrastructure system fails to withstand adverse shocks, it will bring domino hazards to the public [3]. For instance, in 2013, an explosion occurred in Qingdao city of China due to an oil pipeline rupture, resulting in 62 deaths and severe economic losses, about 118,425,000 dollars. Due to multi-round heavy rains in 2020, most cities in southern China, such as Shanghai and Chongqing, suffered from flood disasters causing economic losses of up to 975,664,100 dollars. Obviously, improving urban infrastructure resilience to disasters is a prerequisite for ensuring the normal operations of cities. In recent years, international organizations and some developed countries have begun to use the concept of resilience widely and actively promote resilient infrastructure to improve urban resilience to disasters [4].

Resilience originating from physics described a material's ability to absorb deformation force when deformed by an external force. Later, Holling, an ecologist, applied the concept of resilience to Systems Ecology for the first time, defining it as a measurement of system persistence and ability to absorb changes and disturbances at a system level [5]. Since the 1990s, as the research on resilience had gradually expanded from ecology to other disciplines, the concept of resilience had also been enriched. The multidisciplinary Centre for Earthquake Engineering (MCEER) defined resilience as the system's ability to reduce the possibility of the shocks, absorb vibration and quickly recover afterwards [6]. From the system and information engineering, resilience refers to the ability to withstand severe damage within acceptable degradation parameters, and recover within a reasonable time [7]. The definition of resilience has not been unified. In contrast, three resilience characteristics (i.e., resistance, absorption, and recovery) proposed by Davidson-Hunt [8] were universally approved, laying a foundation for evaluating resilience systems.

As the application of resilience continued to extend to many fields, a series of concepts had been proposed successively, such as ecological resilience [9,10], engineering resilience [11,12], urban resilience [13–15] and infrastructure resilience [16,17]. The subsystems of UI play various roles in the emergency phase, resettlement phase, recovery phase, and reconstruction phase in the risk and are dependent on each other to varying degrees, constituting the overall resilience of urban infrastructure. Most scholars interpreted urban infrastructure resilience from resilience's three characteristics (i.e., resistance, absorption, and recovery) such as Omer, M et al. [18], Jackson, S et al. [19], Bruneau, M et al. [20]. RB Huston [21] referred to urban infrastructure resilience as the joint ability to resist (prevent and endure) any possible harm, absorb initial damage and resume routine operations. In other words, the effectiveness of resilient urban infrastructure could be determined by its ability to predict, absorb, adapt and quickly recover from potentially destructive events [22]. Among them, absorptive capacity was the system's ability to bear damage without significantly deviating from the normal operating performance [23]; adaptability was the system's ability to adapt to shocks under normal operating conditions; recoverability referred to the system's ability to recover quickly from potentially destructive events at low cost. In this paper, urban infrastructure resilience was interpreted as the ability to withstand disasters, absorb losses, and return to normal conditions when disasters occur.

Urban infrastructure development, an extremely complicated process, was challenged by multiple dynamic factors, such as population growth, resource constraints, urbanization, globalization, and climate change, failing to match with the local economic and environmental developments in recent years [24]. With the continuous increase of potential internal and external risks, city administrations had gradually converted from passive response to active risk control. Existing researches on urban infrastructure resilience assessment could be mainly divided into the following two categories. The first category measured the resilience of a single infrastructure. Some scholars primarily focused on the length of post-disaster recovery time to evaluate resilience. Cimellaro, GP et al. [25] constructed

the resilience-time curves from disaster to recovery and pointed out that shortening the recovery time was the key to improving infrastructure resilience. Scott Jackson [19] and Bruneau, M [20] evaluated seismic resilience of communities and physical resilience of infrastructure systems, respectively, based on the resilience-time curves. Infrastructure resilience was determined by the recovery time and affected by other factors (i.e., element recognition, vulnerability analysis, target-setting for resilience, decision-makers cognition, resilience capacity). Francis, R [26] added two important evaluation factors: the possibility of failure and consequences of failure in the system based on recovery time. However, it is worth noting that the quantification of system resilience should also consider the recovery cost, not just the recovery time [21]. Omer, m et al. [18] directly took the ratio of post-disaster transmission value to pre-disaster transmission value of power grid as an index to evaluate infrastructure resilience levels. On this basis, Radvanovsky [21] measured the resilience in critical infrastructure systems on the premise of reducing the investment cost of disaster prevention. Others evaluated the infrastructure resilience comprehensively by constructing an indicator system, such as transportation [27,28], urban drainage system [29], groundwater [24], energy system [30].

However, urban infrastructure resilience is a complex and comprehensive concept whose evaluation process should be abided by the systematicity of infrastructure and dynamics of responding to risks. Most scholars deemed the overall urban infrastructure system the research subject for comprehensive evaluation. Constructing an index system from three benefits of urban infrastructure [31] (i.e., economic benefits, social benefits, environmental effects) or composition characteristics of urban Infrastructure infrastructure [32,33] assess its resilience levels. Besides, some scholars assessed resilience from the perspective of the interaction between infrastructure and external environment, such as infrastructure-environment [34], infrastructure-economy [35], infrastructure-economy-society-environment [36]. It can be found that most indexes in existing evaluation systems of urban infrastructure resilience commonly described the statefulness while ignoring other stages, namely pressure and response. Therefore, these indexes system hardly reflected the dynamic nature of urban infrastructure resilience. So, this paper introduced the adaptive PSR framework into the evaluation of urban infrastructure resilience.

Pressure-State-Response (PSR) was a causal-oriented framework proposed by the Organization for Economic Cooperation and Development (OECD) [37]. The PSR model included three indicators (i.e., pressure, state, and response), as shown in Figure 1. Pressure describes the threats and disturbances caused by the internal and external environment to the UI, explaining why the system changed. State represented the state of the UI under pressure. The response is the self-regulation of UI to adapt to changes and the preventive measures taken by the government and residents [38]. It is not difficult to find that the PSR model connects the causes, impacts, and response to environmental change. The three indicator layers' mutual restriction and effect derived a cycle development network, which continuously adjusted the system to a balanced and stable state. Given its logic, flexibility and comprehensive-ness, the PSR model was widely applied for ecological security assessment [39–42], urban carrying capacity assessment [43,44], and resilience assessment under various disaster risks [45]. UI resilience is dynamic and procedural, and it will also experience the dynamic development process of pre-disaster, mid-disaster and post-disaster states after being impact-ed or disturbed by external. Although most scholars have focused on the dynamic and procedural of resilience, the existing infrastructure resilience assessment process still paid too much attention to the state indicators of UI, ignoring the resilience process and positive feedback process of disturbance and UI. The logic structure of PSR, "cause-effect-response", could make up for the deficiency of the existing UI assessment system to further highlight the dynamic and procedural of resilience. Therefore, the PSR model was introduced in this paper, and construct the UI resilience evaluation system from the input of pressure, the change of UI state under pressure, the autologous feedback of UI and human actions.

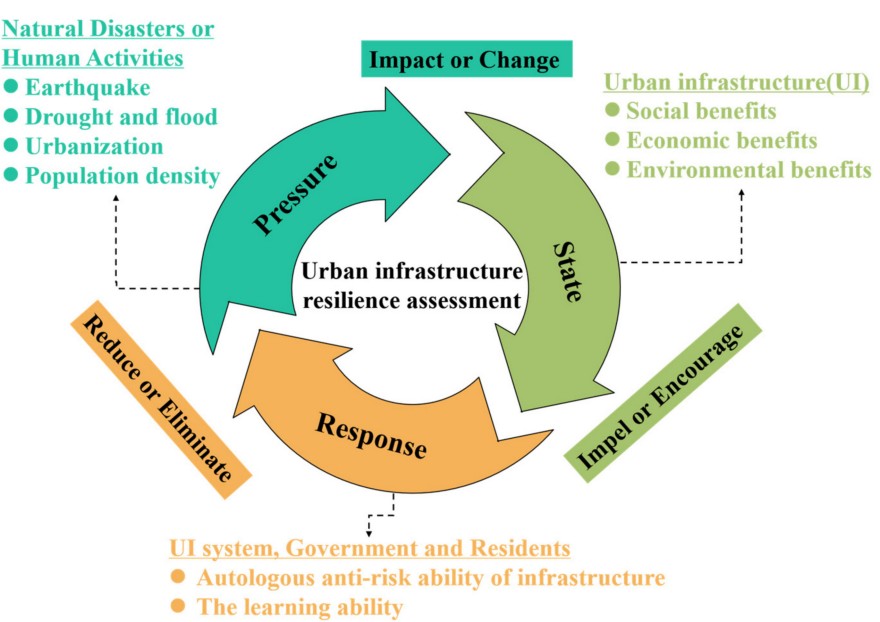

**Figure 1.** PSR Model.

## 2. Materials and Methods

### 2.1. Studied Regions

Four municipalities in China were selected as research objects, namely Beijing, Tianjin, Shanghai, and Chongqing. They are on a large scale and among the most developed cities in China. They have dense populations and a higher demand for urban Infrastructure. As the leaders of three economic circles (Bohai Economic Rim, the Yangtze River Delta Economic Circle and Upper Yangtze River Economic Circle), these municipalities play an increasingly important role in radiating to the surrounding areas and their infrastructure resilience level directly affects the regional development. Therefore, it is necessary to explore the UI resilience level of China by taking four municipalities as examples. Nevertheless, it was worth noting that these four cities may differ in their resilience due to their different development processes and strategic arrangements. Therefore, it was assumed that the resilience levels of different cities are various. This paper attempted to evaluate the pressure, state, response, and infrastructure resilience levels of the four cities based on the PSR framework. The data were from the China Urban Statistical Yearbook, China City Construction Statistical Yearbook, China Environment Statistical Yearbook, and Environmental Quality Bulletin in this study. Mean values of adjacent indicators replace missing data.

### 2.2. Determine the Weight of Each Indicator

There are two main methods to provide weights to indicators: subjective and objective. The subjective approach [46] emphasized the subjective judgment and decision-makers intention and assigned weights on subjective information of decision-makers, such as expert investigation method, analytic hierarchy process (AHP). The results of the subjective approach conform to the subjective wishes of decision-makers, ignoring the information inherent in the data. However, there is no unified standard for the evaluation index system of UI, so the weight calculated by the method of personal preference tends to errors. The objective approach [47] calculated indicator weights on objective mathematical theories, fully reflecting the information in the data. The weight information originating from the indicator itself was determined by the roles of indicators in decision-making.

With information entropy as the core, the entropy method was an objective weighting method to determine the index weights by considering the relationship between the degree changes of indicators and information, and it is widely used in ecological resilience [48,49], urban resilience [50], and risk assessment [51]. The smaller the entropy, the greater the utilization information provided by this parameter. So, entropy could measure the relative

importance of various factors. In fact, under the same evaluation index system, more unstable indicators should be given higher weight to attract the government's attention to improve the resilience level of backward cities, consistent with the principle of entropy weight method. The smaller the variation amplitude of the index, the less information contained in the index, the smaller the effect on the comprehensive evaluation, and the lower the weight value. Therefore, it was assumed that the weights of different indicators are different, and the greater the dispersion degree, the greater the weight of indicators. The entropy weight method was then applied to determine each index weight in this paper.

Step 1: establishing matrix $X$.

Assuming that the evaluation region is divided into $n$ sub-regions, and $m$ means the number of evaluation indicators. So, the dataset ($X$) associated with the evaluation area is expressed as follows:

$$X = \begin{pmatrix} x_{11} & x_{12} & \cdots & x_{1m} \\ x_{21} & x_{22} & \cdots & x_{2m} \\ \vdots & \vdots & \vdots & \vdots \\ x_{n1} & x_{n2} & \cdots & x_{nm} \end{pmatrix}, \tag{1}$$

where, $i = 1, 2, \ldots, n$ and $j = 1, 2, \ldots, m$, and $x_{ij}$ refers to the value of area $i$ relative to indicator $j$.

Step 2: normalize the raw data.

Since units of measurement of each index are various, it projects the original data to the standardized dimensionless values in the interval [0, 1] by the maximum-minimum method, shown in formula (2)–(3).

Where $r_{ij}$ is the normalized value. The closer $r_{ij}$ approaches 1, the higher the resilience, while $r_{ij}$ closer to 0 means lower resilience. Notably, this projection is based on the positive or negative contribution of indicators to the overall resilience of UI. The positive indicators generate positive contributions to enhance resilience, while negative indicators generate negative contributions to inhibit resilience. The process is as follows:

For positive indicator:

$$r_{ij}^{+} = (x_{ij} - \min\{x_j\}) / (\max\{x_j\} - \min\{x_j\}), \tag{2}$$

While, for negative indicator:

$$r_{ij}^{-} = (\max\{x_j\} - x_{ij}) / (\max\{x_j\} - \min\{x_j\}), \tag{3}$$

where $\max\{x_j\}$ and $\min\{x_j\}$ indicate the maximum and minimum values of the index among all evaluation objects, respectively.

Step 3: the entropy of each indicator ($H_j$) is calculated.

$$H_j = -\sum_{i=1}^{n} \left( r_{ij} / \sum_{i=1}^{n} r_{ij} \right) \ln \left( r_{ij} / \sum_{i=1}^{n} r_{ij} \right) / \ln(m), \tag{4}$$

Step 4: the weight of evaluation indicators ($\omega_j$) is calculated. The smaller the entropy value is, the greater $\omega_j$ is, indicating that the index is more important.

$$\omega_j = (1 - H_j) / \left( n - \sum_{j=1}^{m} H_j \right), \tag{5}$$

### 2.3. Three-Stage Resilience Level Assessment

The evaluation results of three stages (pressure, state and response) were respectively calculated in this study on the PSR framework, shown in formula (5). The higher the evaluation result of the stress index ($U_{prssure}$), being faced with minor infrastructure risk and crisis; the higher the evaluation result of the state index ($U_{state}$), the healthier the state;

the higher the evaluation result of response index ($U_{response}$), the timelier the response; and the healthier the infrastructure system.

$$U_{pressure/state/response} = \sum_{j=1}^{m} (\omega_j \times r_{ij}),\tag{6}$$

where, $m$ indicates the number of indicators in each stage (i.e., pressure, state, response) and $r_{ij}$ is the normalized value in matrix $X$. The weighted model was the most common method for evaluating the resilience levels on the PSR due to its simple operation. Specific calculations are as follows:

$$R = W_p U_{pressure} + W_S U_{state} + W_R U_{response},\tag{7}$$

$$W_i = \sum_{j=1}^{k_i} \omega_j,\tag{8}$$

where $k_j$ is the number of assessment indicators in criterion $j$. $W_i$ represents the weight of the stage $i$ ($W_p$ for pressure, $W_S$ for state, and $W_R$ for response), and $R$ is the urban infrastructure resilience level based on PSR.

　　Though the weighted model has been extensively used in resilience assessment, it is noteworthy that it tends to sum the evaluation results of each stage. No matter which criterion layer the evaluation index was placed on, it barely affected the final comprehensive resilience levels. Therefore, it fails to display the coordination degree of the three stages since it cannot effectively reflect the causal logic of the PSR model, and the evaluation result may mislead the judgment.

　　Post-disaster resilience was one of the core factors that give prominence to the concept of disaster resilience of UI and the primary criterion for measuring resilience [26]. Previously, disaster preparedness planning focused on the prevention of destructive events. This strategy may not be sufficient to resist destructive events, especially anti-normal destructive events [52]. In practice, financial constraints make it impossible to strengthen the resilience level of the infrastructure system at all stages to resist all types of destructive events. So, When UI is under pressure, in the current state, the stronger the recovery, the higher the resilience, as shown in Equation (9). Therefore, it is evident that the larger the $R^*$, the higher the resilience of the urban Infrastructure. Meanwhile, based on Maurya et al. [38] and Wei Yang et al. [53], the $U_{preesure/state/response}$ and $R^*$ were divided into five stages by the Non-equidistant division method in this study, as show in Table 1.

$$R^* = \frac{U_{reponse}}{U_{pressure} + U_{state}},\tag{9}$$

**Table 1.** Classification of urban infrastructure resilience levels.

| Category | [0, 0.3) | [0.3, 0.5] | [0.5, 0.7) | [0.7, 0.8) | [0.8, 1] |
|---|---|---|---|---|---|
| $U_{pressure}$ | Serious | High | Moderate | Slight | Minor |
| $U_{state}$ | Damaged | Fragile | Moderately healthy | Healthy | Very healthy |
| $U_{response}$ | No response | Slight response | Moderate response | Somewhat positive | Strong response |
| $R^*$ | No resilience | low resilience | Medium resilience | Higher resilience | Highest resilience |

## 3. Urban Infrastructure Resilience Evaluation Index

　　Constructing a scientific and reasonable evaluation index system was the fundamental premise for evaluating the urban infrastructure system. Based on the PSR model, a comprehensive evaluation index system of urban Infrastructure was established, combined with the complexity, dynamics and openness of the infrastructure system.

### 3.1. Index Selection of Pressure Layer

We mainly considered the pressure layers from natural pressure and artificial pressure. Natural pressure included earthquakes, floods and other natural disasters. Many natural disasters occur in cities, such as earthquake-induced geological hazards, extreme meteorological disasters, drought and flood, lightning disasters, environmental disasters, etc. For UI, four major natural disasters had the most severe impacts on UI and frequently occurred in cities, i.e., earthquakes, floods, fires, and wars. Given data availability, the equivalent magnitude of near-source earthquakes for city and torrential rain days were selected as indicators. Secondly, global warming and frequent extreme weather events posed severe challenges to urban infrastructure; therefore, the extremely hot weather and days above strong gale were added into the element layer of natural pressure. It was worth pointing out that the data of "equivalent magnitude of near-source earthquakes for city and annual rainfalls" were mainly adopted from the classification results of Xu Wei et al. [54]. Human pressure represented the human activities' interference on urban infrastructure, including social progress, economic development, demographic conditions, etc. Therefore, human pressure was constituted by five indicators, including population density, urbanization rate, the total amount of urban sewage, etc. The pressure resilience demonstrated the burden of urban infrastructure caused by natural and human factors. The greater the pressure resilience, the greater the pressure load the urban infrastructure bearded, and the weaker the ability to cope with internal and external disturbances, and vice versa. Therefore, all indicators in the stress stage were negative.

### 3.2. Index Selection of State Layer

The investment, construction and operation of infrastructure had durable impacts on urban social and economic development and environmental resources [55]. Therefore, the state of UI was evaluated from three perspectives: society, economy, and environmental resource. First of all, the social benefits of urban infrastructure demonstrated the active role of infrastructure in promoting urban social progress. Urban infrastructure primarily was in the form of public facilities with main functional services for urban residents, continuously contributing to meeting modern life's ever-increasing demands. Functional urban infrastructure positively impacted people's living standards and social progress [56]. Hence, per capita area of paved roads, the number of public vehicles per 10,000 persons, water coverage rate, and gas coverage rate were selected to characterize the urban infrastructure's social benefits. Besides, infrastructure would have lasting impacts on urban economic development after completion. In the economic state, urban infrastructure not only should reflect its long-term economic benefits but the ability to reduce accident losses after completion. The disaster mitigation capabilities of urban infrastructure characterize by two indicators of "losses converted into cash by traffic accidents and fires". Furthermore, the density of drainpipe in the built-up area and the length of the highway were used to characterize the long-term effectiveness of infrastructure. Finally, the environmental effects in urban infrastructure could alleviate pressure on the ecological environment and supply the local resources environment by daily savings. Indicators were selected to characterize the environmental benefits of infrastructure from the circumstances of resource consumption and possession of urban residents, such as urban green space per capita, water resources per capita, etc.

### 3.3. Index Selection of Response Layer

Response, a positive effect process, included effective measures and countermeasures taken by system subjects in the occurrence and development stages of disturbance, including the ability to recover from disasters and reflective learning in disaster experience [57]. When disturbed by internal and external forces, urban infrastructure would recover from the disturbance with its own capabilities. Besides, the government and urban residents took measures to restore its original state to ensure the normal operation of infrastructure and draw lessons from the disturbance. Therefore, total wastewater discharged, harmless treatment rate of domestic waste, the new civil defence area, the ratio of urban infrastructure maintenance and construction funds over the gross domestic product (GDP), and the

number of hospital beds per capita were selected to measure its recovery capacity. The learning ability was divided into three parts: the supportability of existing innovation ability to infrastructure construction, government funding for improving innovation and learning, and urban residents' ability to acquire and learn information in response to disasters. On the whole, the response resilience characterized the ability of urban infrastructure to respond to disaster impacts. The larger the response resilience, the stronger the ability to cope with disaster shocks, implying the losses caused by disasters to urban infrastructure was usually reduced to the minimum. As for the newly added civil air defence engineering area, it was estimated by "annual completed area of building" as the base, according to the Proportion of equipment in relevant documents issued by local governments, such as *Calculation Rules of Civil Air Defence Area Index Combined with Construction Project in Beijing*.

Given the PSR framework's causal logic and the six major systems of urban infrastructure, the indicators system used in resilience assessment of urban infrastructure was composed of nine pressure indicators, twelve state indicators and nine response indicators, shown in Table 2.

**Table 2.** The indicators system used in resilience assessment of urban Infrastructure on PSR model.

| Function Layer | Criterion Layer | Factor Layer | Descriptions | Properties |
|---|---|---|---|---|
| Pressure | Natural pressure | Torrential rain days | Number of days of rainfall above 50mm for 24 h | Negative |
| | | Extremely hot days | Days with maximum temperature above 35 °C | Negative |
| | | The equivalent magnitude of near-source earthquakes for city | Risk of earthquake disaster | Negative |
| | | Days above strong gale | Days with wind speed between 17.2 m/s and 20.7 m/s | Negative |
| | Human pressure | Population density | The degree of population aggregation in limited land | Negative |
| | | Urbanization rate | The degree of population aggregation to cities | Negative |
| | | Total wastewater discharge | Adverse effects of human activities on resources and the environmental system | Negative |
| | | Industrial sulfur dioxide (SO2) emissions | | Negative |
| | | Industrial dust emission | | Negative |
| State | Social benefit | Per capita area of paved roads | Quality of life of urban residents | Positive |
| | | The number of public vehicles per 10,000 persons | The level of public transport available to urban residents | Positive |
| | | Water coverage rate | Living standard of urban residents | Positive |
| | | Gas coverage rate | | Positive |
| | Economic benefit | Losses converted into cash by fires | Reduce disaster(accident) losses | Negative |
| | | Losses converted into cash by traffic accidents | | Negative |
| | | Density of drainpipe density in the built-up area | Long-term effectiveness of Infrastructure | Positive |
| | | Length of highway | | Positive |
| | Resource environmental benefit | Urban green space per capita | Resource possession of urban residents | Positive |
| | | Water resources per capita | | Positive |
| | | Power consumption per capita | Resource consumption of urban residents | Negative |
| | | Gas consumption per capita | | Negative |
| Response | Recovery and adaptability | Sewage treatment rate | Ability to respond to the pressure of resources environment | Positive |
| | | Innocuous treatment rate of living garbage | | Positive |
| | | Newly added civil air defence engineering area | Government's ability to guarantee society | Positive |
| | | The proportion of urban infrastructure maintenance and construction funds to GDP | Post-disaster emergency rescue | Positive |
| | | Hospital beds per 10,000 population | | positive |
| | Learning ability | Mobile phone coverage rate | Ability of urban residents to acquire and learn information | Positive |
| | | Internet coverage rate | | Positive |
| | | The ratio of intramural expenditure on research and development (R&D) and GDP | Government investment in innovation and learning ability | Positive |
| | | R&D personnel | Supportability of existing innovation ability to infrastructure construction | Positive |

## 4. Discussion

### 4.1. Analysis of Indicator Weights

The entropy weight method was used to determine each indicator weight in the UI resilience evaluation system, as shown in Figure 2a. According to the results of our weight analysis, the proportion of the state layer was the largest at 38.73%, and the response layer and pressure layer accounted for 31.36% and 29.91%, respectively. These results showed that the objective risk caused by the pressure from external was inevitable; meanwhile, the keys to enhancing infrastructure resilience were to improve the stiffness under pressure, recovery and adaptability in the system. In the pressure layer, it can be seen that the potential risk caused by human pressure (with a weight of 17.82%) was far beyond the stimulation of the natural environment (12.09%). Under human pressure, the fan-shaped area angle corresponding to total sewage discharge and urbanization rate is significantly larger than other indicators, demonstrating that both are more dangerous factors from human activities to UI. Therefore, the impact of chronic stress caused by human activities on infrastructure should be paid full attention. In the natural environment, once the disaster caused by the earthquake occurs, the destruction to UI is also severe, embodied by the relative importance of the urban near-source earthquake with 6.83%.

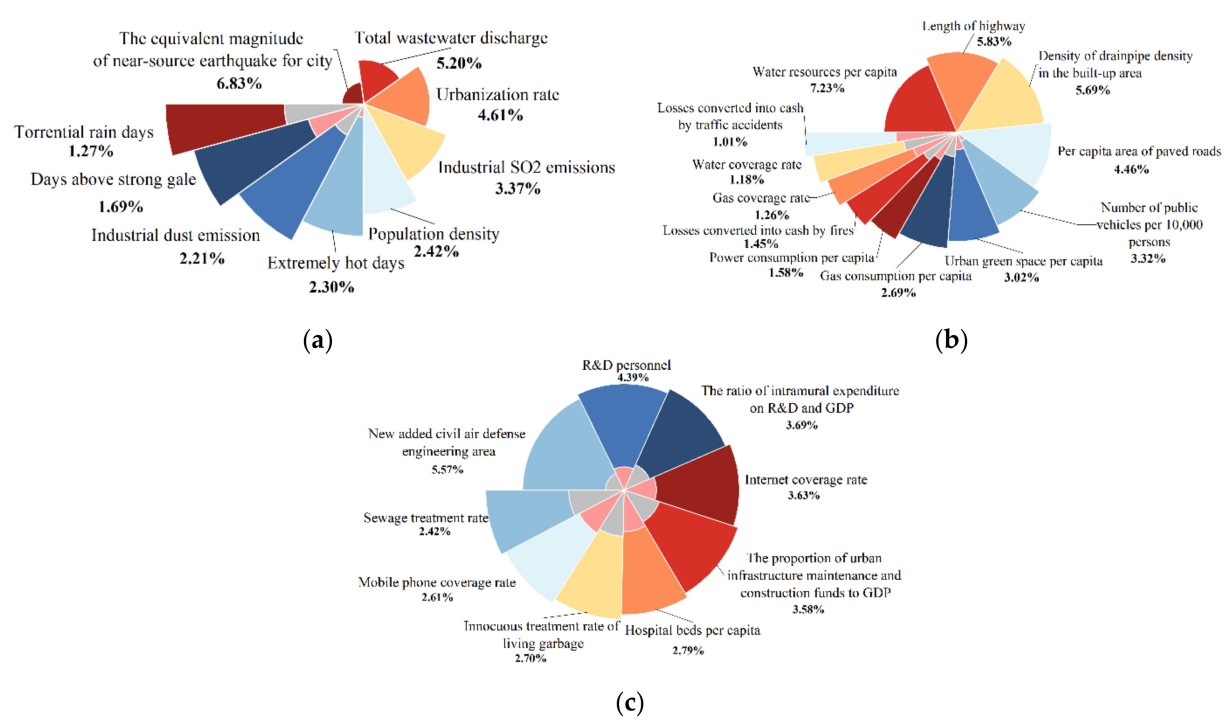

**Figure 2.** Weight proportions of function layer and criterion layer in the evaluation system. (**a**) Weight of pressure layer index; (**b**) Weight of state layer index; (**c**) Weight of response layer index.

At the state stage, the weight of the resource environmental (14.52%) and economic benefits (13.98%) far exceeded the social benefits (10.23%) in infrastructures, shown in Figure 2b. It suggested that strengthening UI's economic and resource environmental benefits is the best approach to improve resilience at the state stage. in resource environmental benefits of UI, water resources per capita was the most significant, accounting for 7.23%. Meanwhile, the length of highway and density of drainpipe density in the built-up area were obtained with the central angle in the economic benefits (shown in Figure 2b), with weights of 5.83% and 5.69%, respectively. However, the social benefits of infrastructure only accounted for 10.23%. Among them, the per capita area of paved roads and the number of public vehicles per 10,000 persons were prominent, with the weights adding up to 7.79%, while the relative importance of other indicators was low. It was primarily since water supply, and gas coverage has almost achieved complete coverage in most cities, well

verified by the original data, that is, water supply coverage and gas coverage have reached 100%. Therefore, it mainly starts with im-proving the road environment to enhance social benefits in UI, such as the per capita area of paved roads, the number of public vehicles per 10,000 persons.

As for the response layer, recovery adaptability accounted for about 17.05%, with maximum impact on infrastructure's ability to resist risks. Located in the same function layer, the learning ability accounted for only 14.31%, indicating it was more vital to improve infrastructure's disaster adaptability than the ability of disaster relief artificially. From a single indicator, the top weights of indicators were newly added civil air defence engineering area with 5.57%, indicating that disaster avoidance was the essential factor for improving urban infrastructure's ability to respond to risks. Moreover, the ratio of intramural expenditure on R&D and GDP weighted with 4.39%, and R&D personnel weighted with 3.69%. It indicated that government investment in innovation and the existing innovation is also a critical factor in improving response-ability.

### 4.2. Changes in the Evaluation Results of Three Stages in Urban Infrastructure

4.2.1. Pressure

During the survey period, the average assessment result of stress **was** the highest in Shanghai (0.476), followed by Tianjin (0.475), Chongqing (0.458) and Beijing (0.443), indicating that the four regions were under high pressure, as shown in Table A2. In Figure 3, the pressure resilience of infrastructure in Beijing constantly fluctuated around 0.450 at the lowest level, indicating internal and external disturbances were relatively active in Beijing with the highest risk coefficient. Beijing, a city located in the seismic belt in geological structure, had relatively high potential risks in the natural environment. Meanwhile, human interference was the most intensive, characterized by the urbanization rate and total wastewater discharge at the forefront of the research cities.

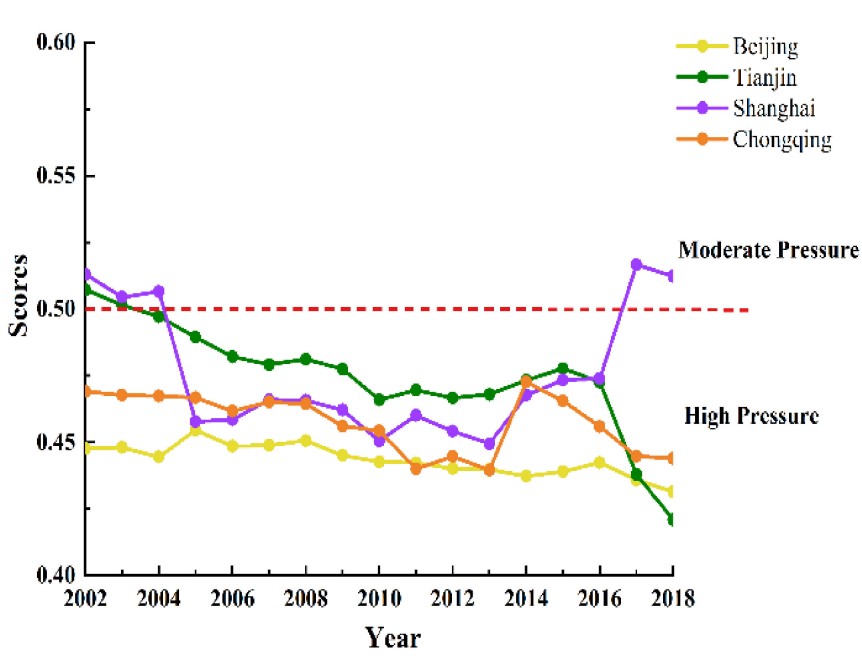

**Figure 3.** Trends in pressure resilience from 2002 to 2018.

Regarding Shanghai, the development levels of pressure resilience were the most unstable during the study period, divided into two stages. From 2002 to 2013, pressure resilience dropped from moderate pressure to high pressure in Shanghai, primarily due to the superimposed consequences of an increased probability of extremely hot days and the excessively rapid human agglomeration. With the continuous adjustment of the industrial structure and the rapid development of industries such as finance and real estate,

not only has it attracted a large number of high-end talents for employment, but it has also provided a **vast** market for some manual workers. Urban population density had doubled in 17 years, causing a surge in disturbances to Shanghai's infrastructure. After 2013, its pressure resilience rebounded to moderate pressure. This was mainly attributed to Shanghai's strict control of population size in recent years, realizing the population changed from a net inflow to a net outflow. As a result, the interference of human activities in cities had gradually weakened, primarily environmental resources, characterized by a reduction of more than 45% in total wastewater, industrial SO2 emissions and industrial dust emissions compared with 2013.

Regarding the magnitude of change, Tianjin's pressure resilience varied the most from 0.507 to 0.421, deteriorating to high pressure from moderate pressure. This was mainly because concentrated high energy-consuming and high-pollution industries, such as steel and petrochemical, incredibly pressured infrastructure capacity to absorb pollutants. In the past 17 years, its total industrial production value showed a soared trend, while the level of urban pressure resilience was limited by the massive discharge of pollutants. The specific characteristics were: the total wastewater discharged and industrial $SO_2$ emissions increased by 2.3 times and 24.70% compared with 2002, respectively. For Chongqing, the development of pressure resilience was primarily limited by the massive emissions of air pollutants and climate conditions of annual high-temperature and rainstorms determined by terrain conditions. So, the pressure resilience fluctuated in the High-Pressure stage throughout the 17 years and showed an apparent downward trend at the end.

### 4.2.2. States

In Figure 4, only Chongqing had relatively high levels of state resilience development, with an apparent upward trend and leading ahead, while the others fluctuated steadily around 0.500. In other words, the overall state of infrastructure in China's municipalities all showed moderate health. Over the years, Chongqing was strived to build a comprehensive transportation hub in southwest China. Its transportation infrastructure had been continuously improved, manifesting that the number of public vehicles per 10,000 persons doubled in 2018 compared with the initial stage of the study. Besides, urban green space per capita also had increased at a rate of 0.14 square meters per year in 17 years. Thus, the evaluation results in Chongqing leapt from 0.518 in 2002 to 0.648 in 2018, with an obvious upward trend, shown in Table A3. Nevertheless, it was worth noting that there is still a significant gap in Chongqing compared with other municipalities, embodied by the inadequate public transport facilities and the incomplete coverage of water and gas. Incredibly, the number of public vehicles per 10,000 persons was far below the average level of municipalities.

There was no significant difference in state resilience development among the remaining cities, as they were all at a moderate health state. The state resilience of infrastructure in Beijing and Shanghai was relatively vulnerable, only about 0.520, at a low level. As political and economic centres in China, Beijing and Shanghai had attracted a large population inflow. Over the past 17 years, the permanent population gross increased by 46.43% on average compared to 2000, well above the pace of infrastructure construction and improvement. As a result, the per capita indicators of infrastructure in these two cities had been in a low state, even a negative growth phenomenon, such as per capita area of paved roads. Secondly, the loss effect spread more widely after the disaster due to mass building density and more high-rise buildings in Beijing and Shanghai, especially in Shanghai. Compared with the initial stage, the losses converted into cash by fires in 2018 almost quadrupled, severely restricting the improvement of state resilience in Shanghai and increasing the possibility of its deterioration to the fragile state. Per capita area of paved roads and density of drainage pipe network in the built-up area in Tianjin were the highest among the four municipalities. Meanwhile, its ability to resist fire was also prominent, which was the main reason why it was superior to Beijing and Shanghai in state resilience. However, it was constrained by massive energy consumption, manifesting per capita consumption far exceeding the 66% average of the four municipalities above 66%.

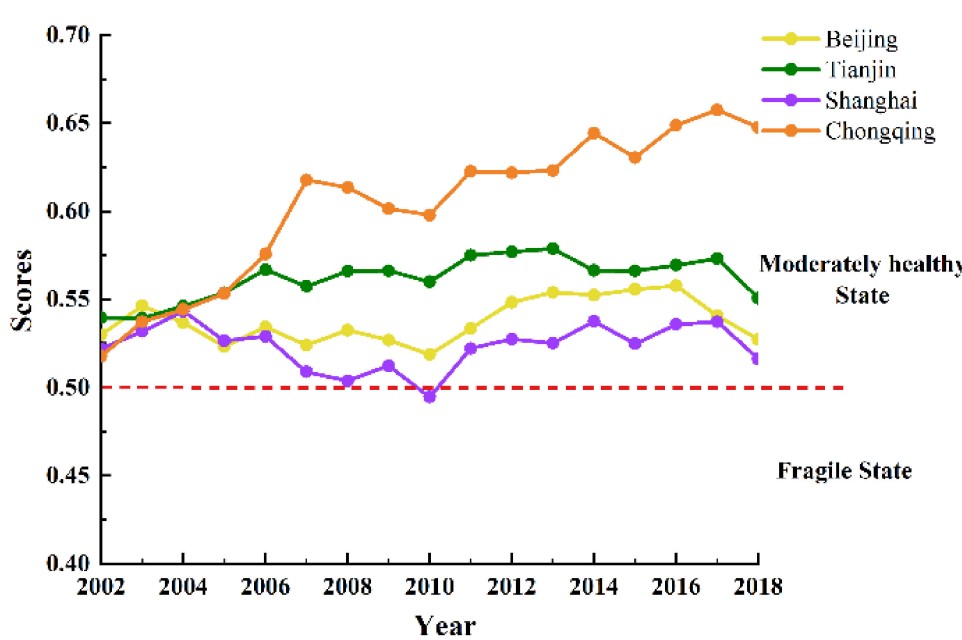

**Figure 4.** Trends in state resilience from 2002 to 2018.

### 4.2.3. Responses

Compared to the previous two stages, the response resilience of infrastructure in four cities varied within relatively noticeable differences, shown in Figure 5 and Table A4. On average, Chongqing had the highest response resilience score at 0.506, the only city in the moderate response stage. High-level civil air defence engineering construction and considerable investment in urban infrastructure maintenance have played a positive or vital role in the recovery and adaptability of urban infrastructure in Chongqing. Chongqing has continuously strengthened pollution control and construction of water environment in recent years, establishing the environmental monitoring and governance system [58]. Therefore, the sewage treatment rate doubled during this period, and the innocuous treatment rate of living garbage increased at an annual growth rate of 0.18%, both reaching leading national levels in 2018. Furthermore, medical service capabilities were also steadily improving, which manifested that the hospital beds per 10,000 population doubled in 17 years. As a result, response resilience in Chongqing shifted into the moderate response from the slight response.

The development level of infrastructure response in Beijing was in the range of [0.418, 0.459], ranking the latest among the four cities. Beijing's water environment pollution was relatively severe [59]. In recent years, the continuous upgrading of sewage treatment plants and strengthening comprehensive treatment of the water environment has dramatically improved the sewage treatment capacity while still insufficient to absorb pollution. Moreover, the newly added civil air defence engineering area was also far lower than that of other cities, constraining the resilience and adaptability of its infrastructure to some extent.

As for Shanghai, it decreased from 0.503 (Moderate response) in 2002 to 0.478 (Slight response) in 2018, primarily attributed to a gradual decrease in its investment in the operation and maintenance of urban infrastructure, with ended up only three-fifths of the average of all research cities. However, Tianjin had relatively stable development trends in response resilience, with an average response resilience of 0.452. It was found **that** in the case of low infrastructure resilience and adaptability, most of the indicators that characterize learning ability are far inferior to other cities, such as hospital beds per 10,000 population, mobile phone coverage rate, leading to Tianjin with only a slight response at the end of the research period.

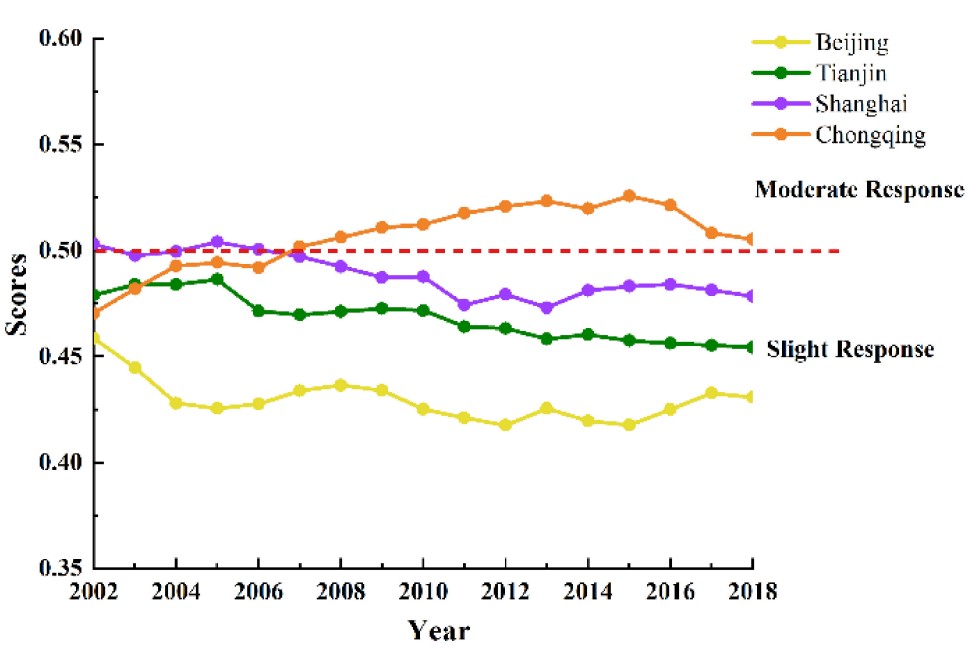

**Figure 5.** Trends in response resilience from 2002 to 2018.

*4.3. Comprehensive Resilience Level of Urban Infrastructure*

Based on the PSR model for UI resilience of four municipalities in China, we found that all cities were continuously at low resilience from 2002 to 2018, even showing a slight downward trend. The result indicated that the overall resilience levels of municipalities' infrastructure in China were generally poor, very likely even not resilient. In Table A5, Shanghai ranked first with average infrastructure resilience scores of 0.489, followed by Chongqing (0.477), Tianjin (0.452) and Beijing (0.424), all classified as low resilience.

Generally speaking, the development trend in Shanghai could be divided into two stages, with 2010 as the boundary, shown in Figure 6. It shifted from low resilience (0.486) in 2002 to medium resilience (0.516) in 2010 while steadily declining until 2018, ultimately to low resilience (0.465). The improvement of the score at the previous stage may appear to have benefited from the descent range of pressure resilience (−0.06) and state resilience (−0.03) more significant than response resilience (−0.02), closely related to the economic restructuring and efficient industrial waste abatement in Shanghai. While from 2010 to 2018, the resilience of pressure and state had been improved to some extent, the response resilience of infrastructure showed a downward trend, thus dragging down Shanghai into low resilience stage. Meanwhile, the resilience development of the other cities was at low resilience with little change.

Regarding Chongqing, the level of infrastructure resilience fluctuated between 0.461 and 0.492. This was mainly due to extremely hot days that occurred more frequently caused by special geographic conditions, resulting in unsatisfactory pressure resilience. Moreover, the state resilience of infrastructure was generally increasing. However, it respectively experienced two large drops in 2010 and 2015, indicating that the anti-interference ability of the infrastructure system was relatively unstable. As a result, Chongqing had always been at low resilience, hardly changing over 17 years.

While the infrastructure resilience in Beijing and Tianjin remained at a below-average level throughout the study period. With the rapid development of the economy, Tianjin, a traditional heavy industry city, severely deteriorated the ecological environment, bringing about the continuous decline of its pressure resilience in the past 17 years. Besides, huge resource consumption, especially power consumption per capita and gas consumption per capita, had become a major problem, restricting the continuous development of environmental resource benefits of infrastructure. The negative effect produced by the two stages of pressure and state far exceeded the positive effect of the response, leading to a

downward trend in Tianjin's infrastructure resilience level. As revealed in Figure 6, Beijing was the worst in terms of infrastructure resilience. As the population soared, potential pressure within the system had been increasingly emerging, well above the upper limit of the existing infrastructure capacity. As a result, the social and environmental resource benefits of infrastructure in Beijing had been negatively increased; simultaneously, the system's learning ability failed to promote timely or even decreased slightly, creating a vicious cycle. In short, Beijing consistently scored the lowest on infrastructure resilience levels, resulting from the uncoordinated development of the three-stage, manifesting increased pressure, deterioration of the state and lag of response capacity.

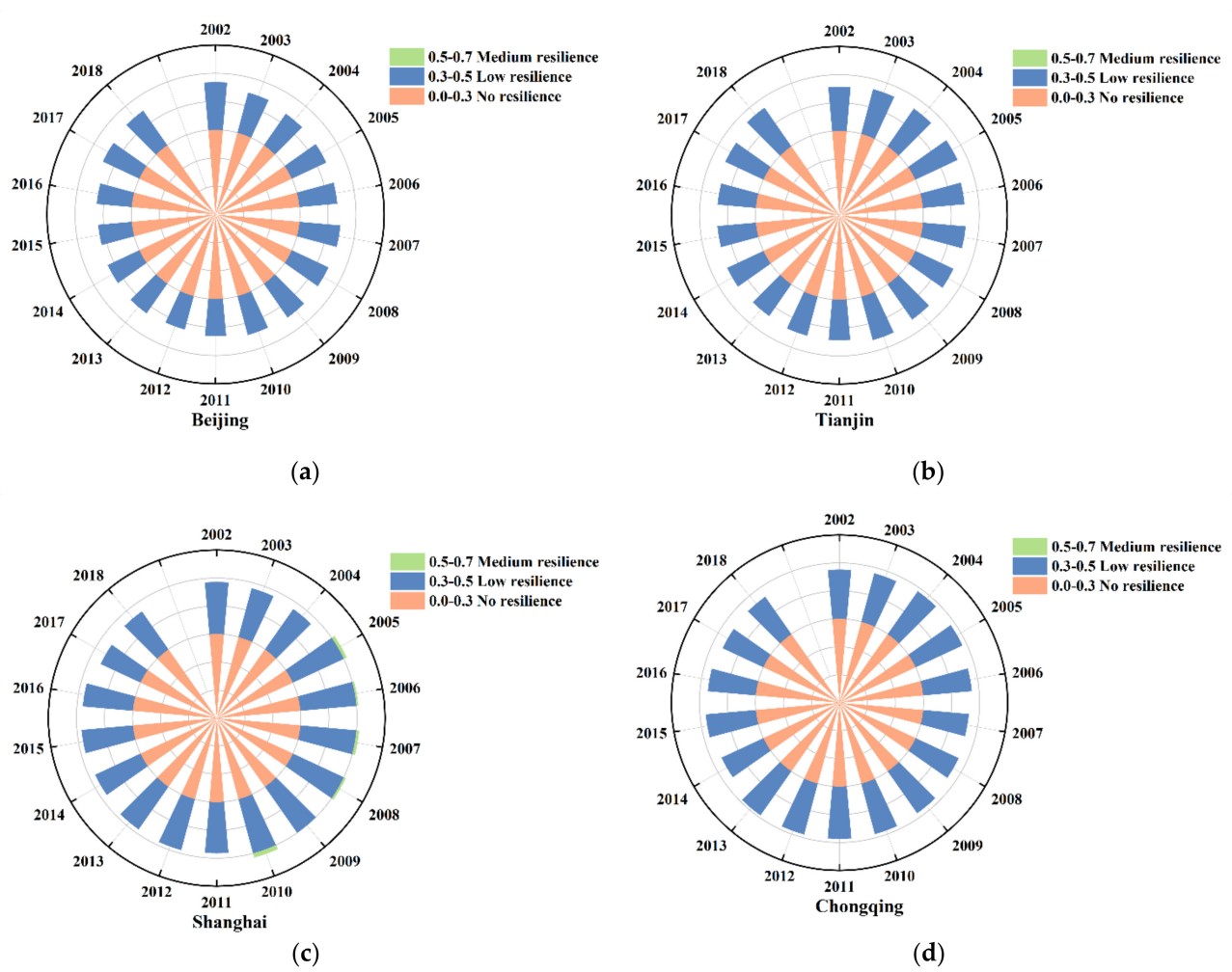

**Figure 6.** The trends of urban infrastructure resilience levels from 2002 to 2018. (**a**) Beijing; (**b**) Tianjin; (**c**) Shanghai; (**d**) Chongqing.

## 5. Conclusions

Pressure-State-Response (PSR) was introduced in this paper; its cause-effect-response logical structure was used to construct a UI resilience assessment system, reflecting dynamic and process characteristics. Moreover, the resilience level of infrastructure in four municipalities in China from 2002 to 2018 is measured and analyzed to explore the long-term temporal changes of UI resilience.

The state layer had the most significant impact on the resilience level of UI with the proportion of 38.73%, especially the state of environmental resources. The benefit of per capita water resources was the most obvious, while it was determined by local nature, leading it challenging to improve state resilience. Subsequently, the weight accounted for a large number of lengths of highway with 5.83%, density of drainpipe density in the built-up

area with 5.69%. Thus, it is necessary to strengthen the construction of public transport facilities to improve the urban traffic environment and enhance the density of drainage networks to ensure the drainage capacity in the city. In the response stage, attention should be paid to the construction of civil air defence to ensure the emergency capacity of urban infrastructure. Compared with the other stag-es, the pressure layer **was** relatively less important, while seismic fortification should be paid the most attention because of the considerable weight, accounting for 6.83%. We should strengthen seismic review and strictly control the construction workman-ship so that the engineering infrastructure could meet the precautionary seismic intensity stipulated by the countries.

Overall, the levels of UI resilience were poor in these four municipalities, continuously at the low level throughout the study period. However, there were still some differences among the four cities. Based on this, we put forward corresponding measures to improve the resilience in different cities. Although Shanghai ranked the highest, its large population base has led to negative growth in per capita infrastructure ownership. Meanwhile, the expansion of the fire loss effect caused by excessive population density resulted in state resilience in Shanghai being the lowest in the four municipalities. So, for Shanghai, it should minimize the disturbance of human activities on UI as possible and gradually shift to the development of urban agglomerations by eliminating the central effect of cities. The comprehensive resilience of Chongqing was followed by Shanghai, mainly due to severe air pollution in its pressure resilience and the water supply and gas in the state layer without full coverage. Given its unique geographical conditions, Chongqing should pay more attention to controlling the emission of air pollutants and continuously improving the infrastructure of people's livelihood, to shorten the gap with other municipalities. The comprehensive resilience in Tianjin ranking was backward, due to its poor response capacity, such as insufficient hospital beds, and limited disaster acquisition channels. In the process of UI construction and development, Tianjin should prioritize strengthening urban emergency response capacity. Beijing had the lowest comprehensive resilience. The reason was that the potential risks of the natural environment were relatively prominent, resulting in low-pressure resilience, especially in low response capacity. Reflected in the fact that the sewage treatment capacity cannot keep up with the speed of urban development, and the new civil air defence area is far lower than that of other municipalities. Therefore, compared with other municipalities, it **was** most urgent for Beijing to improve its infra-structure adaptation and recovery capacity.

There were large differences in resilience development levels among the three stages of pressure, state, and response, manifested by a large improvement in state resilience, decreased pressure resilience, while the response resilience remained unchanged in the fluctuation. In other words, the uncoordinated development level of three stages in four cities was also a major reason for low resilience, especially in stages of pressure and state. The change of a certain pressure factor or state factor would affect the overall structure of the urban infrastructure system, thus forming a new state-response relationship: a new cycle. Therefore, in constructing resilient infrastructure, full attention should be paid to the coupling and circular relationship among various elements to achieve dynamic evaluation and management. In the three stages, improving the overall resilience of UI from the response capacity is most critical and effective. Overall, cities should pay attention to emergency capacity building, strengthen the technical support of emergency management, and accelerate the application of emerging technologies in urban emergency management, such as accelerating the application of emerging technologies such as big data, cloud computing and artificial intelligence.

**Author Contributions:** Conceptualization, E.W. and Y.W.; methodology, Y.J. and M.C.; software, Y.J.; validation, M.C., J.Z. and Y.W.; formal analysis, Y.J.; data curation, Y.J.; writing—original draft preparation, Y.J. and Y.W.; writing—review and editing, E.W., Y.W., J.Z. and M.C.; visualization, Y.J. All authors have read and agreed to the published version of the manuscript.

**Funding:** This work was supported by the Ministry of Housing Urban-Rural Development of China (2019K014) and the Practice and Innovation Fund for University Students of Jiangsu Province (201910304039Z).

**Conflicts of Interest:** The authors declare no conflict of interest.

## Appendix A

**Table A1.** Weights of indicator in the urban Infrastructure evaluation system.

| Function Layer | Criterion Layer | Factor Layer | Weights (%) |
|---|---|---|---|
| Pressure | Natural pressure | Torrential rain days | 1.27 |
| | | Extremely hot days | 2.30 |
| | | The equivalent magnitude of near-source earthquakes for city | 6.83 |
| | | Days above strong gale | 1.69 |
| | Human pressure | Population density | 2.42 |
| | | Urbanization rate | 4.61 |
| | | Total wastewater discharge | 5.20 |
| | | Industrial SO2 emissions | 3.37 |
| | | Industrial dust emission | 2.21 |
| Sate | Social benefits | Per capita area of paved roads | 4.46 |
| | | The number of public vehicles per 10,000 persons | 3.32 |
| | | Water coverage rate | 1.18 |
| | | Gas coverage rate | 1.26 |
| | Economic benefits | Losses converted into cash by fires | 1.45 |
| | | Losses converted into cash by traffic accidents | 1.01 |
| | | Density of drainpipe density in the built-up area | 5.69 |
| | | Length of highway | 5.83 |
| | Environmental resource benefits | Urban green space per capita | 3.02 |
| | | Water resources per capita | 7.23 |
| | | Power consumption per capita | 1.58 |
| | | Gas consumption per capita | 2.69 |
| | | Sewage treatment rate | 2.42 |
| | | Innocuous treatment rate of living garbage | 2.70 |
| Response | Recovery and adaptability | Newly added civil air defence engineering area | 5.57 |
| | | The proportion of urban infrastructure maintenance and construction funds to GDP | 3.58 |
| | | Hospital beds per 10,000 population | 2.79 |
| | | Mobile phone coverage rate | 2.61 |
| | Learning ability | Internet coverage rate | 3.63 |
| | | The ratio of intramural expenditure on R&D and GDP | 3.69 |
| | | R&D personnel | 4.39 |

**Table A2.** Scores in pressure resiliencelevels from 2002 to 2018.

| Category | Beijing | Tianjin | Shanghai | Chongqing |
|---|---|---|---|---|
| 2002 | 0.448 | 0.507 | 0.513 | 0.469 |
| 2003 | 0.448 | 0.501 | 0.504 | 0.468 |
| 2004 | 0.444 | 0.497 | 0.506 | 0.467 |
| 2005 | 0.455 | 0.489 | 0.458 | 0.467 |
| 2006 | 0.448 | 0.482 | 0.458 | 0.462 |
| 2007 | 0.449 | 0.479 | 0.466 | 0.465 |
| 2008 | 0.451 | 0.481 | 0.466 | 0.464 |
| 2009 | 0.445 | 0.477 | 0.462 | 0.456 |
| 2010 | 0.443 | 0.466 | 0.450 | 0.454 |
| 2011 | 0.442 | 0.470 | 0.460 | 0.440 |
| 2012 | 0.440 | 0.467 | 0.454 | 0.445 |
| 2013 | 0.440 | 0.468 | 0.449 | 0.440 |
| 2014 | 0.437 | 0.473 | 0.468 | 0.473 |
| 2015 | 0.439 | 0.478 | 0.473 | 0.465 |
| 2016 | 0.442 | 0.473 | 0.474 | 0.456 |
| 2017 | 0.436 | 0.438 | 0.517 | 0.445 |
| 2018 | 0.431 | 0.421 | 0.512 | 0.444 |

**Table A3.** Scores in state resilience levels from 2002 to 2018.

| Category | Beijing | Tianjin | Shanghai | Chongqing |
|---|---|---|---|---|
| 2002 | 0.530 | 0.540 | 0.522 | 0.518 |
| 2003 | 0.546 | 0.539 | 0.532 | 0.537 |
| 2004 | 0.537 | 0.546 | 0.543 | 0.544 |
| 2005 | 0.523 | 0.554 | 0.526 | 0.553 |
| 2006 | 0.534 | 0.567 | 0.529 | 0.576 |
| 2007 | 0.524 | 0.557 | 0.509 | 0.618 |
| 2008 | 0.532 | 0.566 | 0.504 | 0.613 |
| 2009 | 0.527 | 0.566 | 0.512 | 0.601 |
| 2010 | 0.519 | 0.560 | 0.495 | 0.598 |
| 2011 | 0.533 | 0.575 | 0.522 | 0.623 |
| 2012 | 0.548 | 0.577 | 0.527 | 0.622 |
| 2013 | 0.554 | 0.579 | 0.525 | 0.623 |
| 2014 | 0.552 | 0.566 | 0.538 | 0.644 |
| 2015 | 0.556 | 0.566 | 0.525 | 0.631 |
| 2016 | 0.558 | 0.569 | 0.536 | 0.649 |
| 2017 | 0.541 | 0.573 | 0.537 | 0.658 |
| 2018 | 0.527 | 0.551 | 0.516 | 0.648 |

**Table A4.** Scores in response resilience levels from 2002 to 2018.

| Category | Beijing | Tianjin | Shanghai | Chongqing |
|---|---|---|---|---|
| 2002 | 0.459 | 0.479 | 0.503 | 0.470 |
| 2003 | 0.445 | 0.484 | 0.498 | 0.482 |
| 2004 | 0.428 | 0.484 | 0.499 | 0.493 |
| 2005 | 0.425 | 0.486 | 0.504 | 0.494 |
| 2006 | 0.428 | 0.471 | 0.500 | 0.492 |
| 2007 | 0.434 | 0.470 | 0.497 | 0.502 |
| 2008 | 0.436 | 0.471 | 0.492 | 0.506 |
| 2009 | 0.434 | 0.473 | 0.487 | 0.511 |
| 2010 | 0.425 | 0.472 | 0.488 | 0.512 |
| 2011 | 0.421 | 0.464 | 0.474 | 0.518 |
| 2012 | 0.418 | 0.463 | 0.479 | 0.521 |
| 2013 | 0.425 | 0.458 | 0.473 | 0.523 |
| 2014 | 0.420 | 0.460 | 0.481 | 0.520 |
| 2015 | 0.418 | 0.457 | 0.483 | 0.526 |
| 2016 | 0.425 | 0.456 | 0.484 | 0.521 |
| 2017 | 0.433 | 0.455 | 0.481 | 0.508 |
| 2018 | 0.431 | 0.454 | 0.478 | 0.505 |

**Table A5.** Urban infrastructure resilience levels from 2002 to 2018.

| Category | Beijing | Tianjin | Shanghai | Chongqing |
|---|---|---|---|---|
| 2002 | 0.469 | 0.458 | 0.486 | 0.477 |
| 2003 | 0.447 | 0.465 | 0.480 | 0.480 |
| 2004 | 0.436 | 0.464 | 0.476 | 0.487 |
| 2005 | 0.435 | 0.466 | 0.512 | 0.485 |
| 2006 | 0.435 | 0.449 | 0.507 | 0.474 |
| 2007 | 0.446 | 0.453 | 0.510 | 0.463 |
| 2008 | 0.444 | 0.450 | 0.508 | 0.470 |
| 2009 | 0.447 | 0.453 | 0.500 | 0.483 |
| 2010 | 0.442 | 0.460 | 0.516 | 0.487 |
| 2011 | 0.432 | 0.444 | 0.483 | 0.487 |
| 2012 | 0.422 | 0.444 | 0.488 | 0.488 |
| 2013 | 0.428 | 0.438 | 0.485 | 0.492 |
| 2014 | 0.424 | 0.443 | 0.479 | 0.465 |
| 2015 | 0.420 | 0.438 | 0.484 | 0.480 |
| 2016 | 0.425 | 0.438 | 0.479 | 0.472 |
| 2017 | 0.443 | 0.450 | 0.457 | 0.461 |
| 2018 | 0.449 | 0.468 | 0.465 | 0.463 |

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
