# Peer review of "Measuring Urban Infrastructure Resilience via Pressure-State-Response Framework in Four Chinese Municipalities"

_applsci, doi:10.3390/app12062819_

Round 1

Reviewer 1 Report

The article addresses an important problem of assessment of resilience, in particular urban infrastructure resilience of four Chinese cities. The article has a good structure. The content is presented in a clear and logical sequence. Methods are correct.

Despite positive assessment of the research problem I suggest introducing the following corrections:

  • no explanations for the abbreviations used, eg AHP (line 180),
  • lines 201 and 204-5 – symbols m,n are not precisely defined, and used in to many meanings,
  • line 231 – a symbol m is explained, which is not presented above,
  • is not clear what “public vehicles” are? Do they belong to institutions own by state? Or to companies? Or to inhabitants?
  • spelling or writing mistakes - eg. line 352, 549, 577

Author Response

Thank you for your comments concerning our manuscript entitled “Measuring Urban Infrastructure Resilience via Pres-sure-State-Response Framework in Four Chinese Municipalities” (applsci-1585845). We have considered all of them and have carried out appropriate revisions, and highlighted in blue. We present below a point-by-point response to the comments.

Reviewer 2 Report

The article addresses a topic of great interest in a complete and exhaustive way, framing well the complexity of the issues involved. The work can make an important contribution to the state of the art in the assessment of existing infrastructures in need of restructuring and related issues as it is clear in the exposition of the methods used and linear in the organization of the weight that each indicator assumes with respect to the vision of integrated system.

In the introduction, the objectives are clear to the reader and the sources cited are up-to-date.

The methodology can help the subjects it addresses and is consistent as well as well explained. In particular, public administrations and their respective consultants can find a valid evaluation tool in the methodology shown.

The discussion is well supported by the results they have obtained and very articulated. The goals they had set have been achieved but I ask the authors not to report the Yuan as a currency, but to transform them into dollars for easier reading to a large audience. Also, could you specify your normalization process better? For example from line 379 to line 488, in order to better follow the reasoning that the authors make in paragraphs 4.2.1, 4.2.2 and 4.2.3 it would be useful to have the values ​​of each indicator multiplied by the respective weight defined in the Entropy Weight Method, represented as in figure 2.a, 2.b, 2.c.

The conclusions are operational and practical with respect to the objectives, and even go so far as to suggest some issues to be particularly monitored within the complex urban system.

In my opinion the work can be published after minor revision.

Minor revision –

-Please specify your normalization process better

-Please do not to report the Yuan as a currency, but to transform them into dollars for easier reading to a large audience.

-Line 379 - after the word "stress" there are two empty brackets. I suppose there goes a reference.

-from line 379 to line 488, in order to better follow the reasoning that the authors make in paragraphs 4.2.1, 4.2.2 and 4.2.3 it would be useful to have the values ​​of each indicator multiplied by the respective weight defined in the Entropy Weight Method, represented as in figure 2.a, 2.b, 2.c.

Author Response

(The authors gave the same response as above.)

Reviewer 3 Report

Overall. The paper is excellent. There are a few improvements to presentation and methodology listed in the attached file. All of my comments do not need to be addressed and some are meant to be for the overall research trajectory and not for this paper. A copy edit is needed for some misspelling, word choices and verb tenses.

Author Response

(The authors gave the same response as above.)
